# A Review on Aerodynamic Characteristics and Wind-Induced Response of Flexible Support Photovoltaic System

**Fubin Chen** [1,2], **Yuzhe Zhu** [2], **Weijia Wang** [2], **Zhenru Shu** [3,*] **and Yi Li** [2]

[1] Key Laboratory of Bridge Engineering Safety Control by Department of Education, Changsha University of Science and Technology, Changsha 410114, China
[2] School of Civil Engineering, Changsha University of Science and Technology, Changsha 410114, China
[3] School of Civil Engineering, Central South University, Changsha 410075, China
[*] Correspondence: shuzhenru@csu.edu.cn

**Abstract:** Photovoltaic (PV) system is an essential part in renewable energy development, which exhibits huge market demand. In comparison with traditional rigid-supported photovoltaic (PV) system, the flexible photovoltaic (PV) system structure is much more vulnerable to wind load. Hence, it is imperative to gain a better understanding of the aerodynamic characteristics and wind-induced response of flexible photovoltaic system. The main objective of this paper is to provide a comprehensive review on the state-of-the-art studies focusing on the aerodynamic characteristics and wind-induced response of flexible PV system. Relevant studies have been carried out, using either physical or numerical simulation tools, and the effect of a series of governing parameters, such as spacing ratio, angle of attack, inclination and position are considered. In addition, dynamic response of these flexible structures, including buffeting, flutter, vortex-induced vibration, are also discussed and documented.

**Keywords:** flexible photovoltaic (PV) system; wind load; aerodynamic characteristics; dynamic response

## 1. Introduction

Actively exploring renewable energy source has long been a focus of attention in both academic and industrial fields, which plays an important role in reducing environment degradation and accelerating sustainable development. In 2020, China was committed to achieving the goal of "reaching the peak of carbon dioxide emissions by 2030 and achieving carbon neutrality by 2060" at the 75th session of the United Nations General Assembly [1], which highlights the importance of harnessing renewable energy in China. As one of the most popular renewable energy sources, solar energy has received widespread attention. This is mainly because the relative simplicity of the electricity production process, the possibility of reducing the global warming effect and air pollution and its inexhaustibility [2,3]. The process of electricity generation for solar energy is non-intrusive, needs little maintenance and can be used at nearly any scales [4]. Moreover, the emerging of solar energy also lies in the trend of continuous decreasing price of solar photovoltaic (PV) panels, whereas the cost of traditional energy sources has been increasing [5,6].

PV modules were initially designed to be installed on the rooftop. However, given the limited space of roof area, ground-mounted PV panels with fixed support structures have been more frequently used. More recently, PV panels and solar power stations have also been positioned under complex terrain conditions, such as mountains, fish ponds and sewage plant. This is mainly because of the scarcity in useable land space. Under the circumstance, the span of the fixed PV supports is too small, which leads to the innovative use of flexible PV module support structure.

The concept of flexible PV support structure was first introduced by Baumgartner [7–9] in which the PV panels were supported by cables (see Figure 1). In comparison with traditional fixed supported PV system structure, the flexible PV system exhibits several

advantages, including a more flexible span length, more rational utilization of land space, lower costs and shorter installation time [10,11]. However, it is also equipped with cable structure bearing (see Figure 2), smaller cable stiffness, lighter weight and lower vibration frequency, which results in more pronounced sensitivity to wind load [12,13]. One of the most important tasks on PV system, either fixed or flexible, is the analysis of aerodynamic loads acting on the solar panels, and indirectly on the support structures [14]. There has been clear evidence showing that wind can cause tremendous damage to PV system [15], as shown in Figure 3. On this account, it is of great practical importance to gain an in-depth understanding of the aerodynamic characteristics and wind-induced response of flexible PV systems. Note that numerous studies have been carried out on this purpose, by means of respectively field measurement, physical and numerical simulation. The remaining contents are structured as follows: Section 2 focuses on the review of the wind load characteristics on flexible PV system structures, Section 3 deals mainly with the wind-induced dynamic responses of flexible PV structures, and Section 4 summarizes the major conclusions.

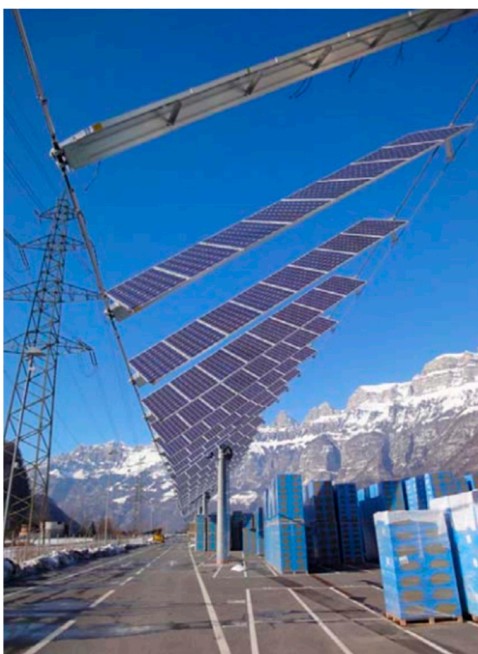

**Figure 1.** Example of the flexible PV system installation [7].

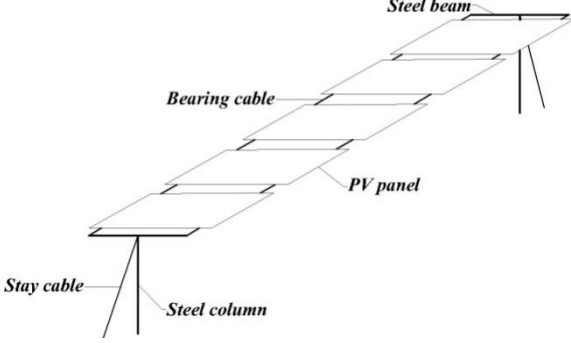

**Figure 2.** Schematic diagram of flexible PV system.

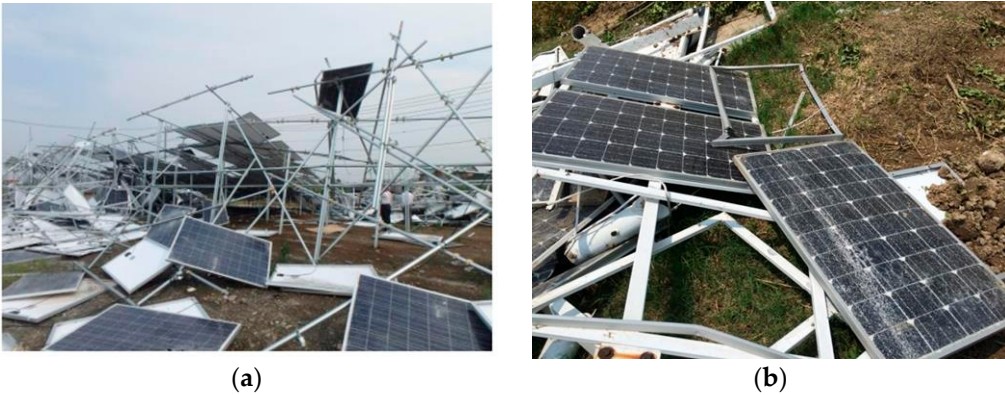

| (**a**) | (**b**) |

**Figure 3.** Example of wind-induced damages on PV panel arrays: (**a**) In Iseisaki city, Gunma prefecture, Japan in 2015; (**b**) In Yancheng city, Jiangsu province, China in 2015.

## 2. Wind Load Characteristics on Flexible PV System

In comparison with rigid, ground-supported PV structure, the inclination of the support for flexible PV is often smaller, which is due partly to the limitation of the structural system [16]. The effect of three-dimensional flow tends to occur in the flow field for a certain gap between PV modules. In addition to the inclination $\beta$, the variation of the angle of attack of $\alpha$ and the PV panel spacing ratio $S$ are also decisive factors regarding the surface wind loads on flexible PV system (see Figure 4). To properly diagnose the effect of these factors, wind tunnel and numerical simulations are commonly used [17–21].

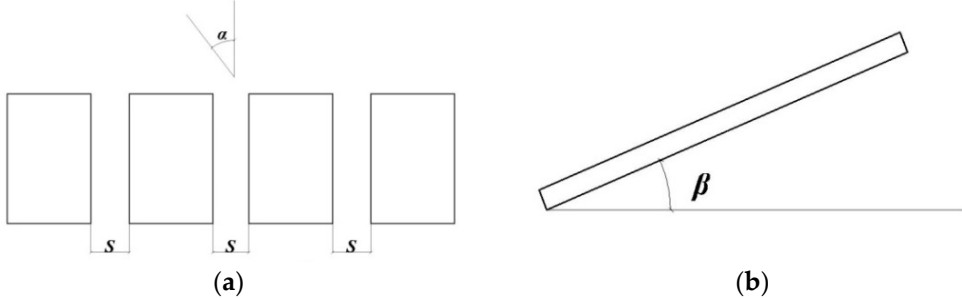

| (**a**) | (**b**) |

**Figure 4.** Definition of governing parameters in PV system: (**a**) spacing ratio and angle of attack; (**b**) inclination.

### 2.1. Wind Tunnel Test

Physical simulation in wind tunnel facility is arguably one of the most widely-used techniques in wind engineering community to diagnose the wind load characteristics on structures [22–24]. With consideration of PV system, He et al. [25] studied the wind-induced response of the flexible PV modules under different wind speeds, finding that due to the characteristics of high flexibility and low damping of PV modules, relatively strong wind-induced vibration would occur when wind speed exceeded the critical value, and the amplitude would magnify with the increase of wind speed, damaging the safety of modules (see Figure 5). Through a rigid model wind tunnel pressure experiment, Du et al. [26] found that under different wind directions, the mean and pulsating wind pressure distribution of long-span flexible PV supports are similar, decreasing for absolute value along with the direction of the inflow, and extreme wind pressure appeared at the corner of the windward leading side under some certain wind directions, and the overall wind pressure tended to expand when the inclination increased from 0° to 10°. Furthermore, Li et al. [27] studied the flutter performance of the flexible PV supports by wind tunnel experiment of elastic suspension segmental models, investigating the influence factor of the flutter stability. Ma et al. [28] carried out a series of wind tunnel tests to examine the connection between wind loads and several governing parameters including inclination, spacing, installation

position, etc. It was shown that for flexible PV structure, the maximum wind load occurs when the wind attacks at an angle of 150° or 180°, and the body shape coefficient is found to increase in a linear manner with the increase of inclination angle. The wind load on PV system tends to vary significantly in accordance to the adjustment of pitch ratio and inclination. The sensitivity of wind load on the leeward side is more pronounced than that on the windward side, while in terms of magnitude, the wind load on the windward side is much larger than that on the leeward side [29]. When the bottom of the PV system was blocked, the maximum wind suction of the entire system appears to increase, while the maximum wind pressure, as well as the bending moment, is found to reduce [30]. Banks [31] characterized the wind loads on tilted PV panels installed on the roofs of flat buildings as a function of shape. The results unveil that the different building shapes can influence the wind loads by modifying the vortex shape of the flow field. It was shown that buildings with larger depth-width ratio can withstand larger wind load than smaller buildings. Regarding the wind-resistant design, the eccentricity of wind load has received much attention. Zhang et al. [32] used different wind speeds to analyze the stress of PV system under 41° of tension, which showed that the wind load point deviates from that of the PV system geometry center, i.e., eccentric distribution. It is worthwhile noting that in certain cases, eccentric distribution of wind load can lead to overturning of the support structure.

(a)

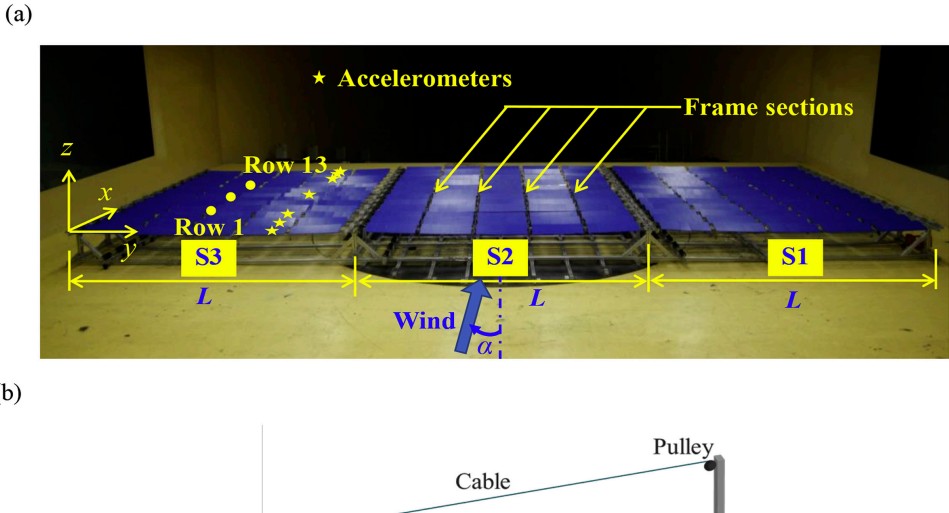

(b)

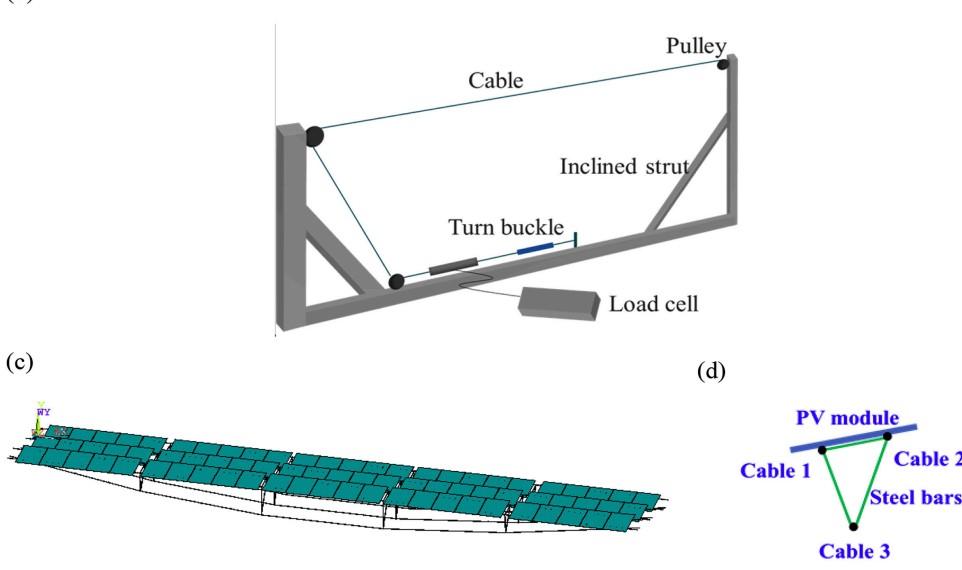

(c)            (d)

**Figure 5.** Example of wind tunnel based study on flexible PV system: (**a**) Three spans and thirteen rows of the test model installed in wind tunnel; (**b**) Sketch of a U-shaped framework; (**c**) One span and three rows of FEM; (**d**) Cross-section of the PV modules [25].

For the arrangement of PV panels, the array is one of the most popular types since it allows better use of the site and higher efficiency of operation. It also allows the use of large-span structure. When the PV panels positioned in an array, the shielding effect on the downstream are inevitable, which are found to be more significant under larger inclination. Evidence has indicated that with larger-enough spacing distance, the wind load of the downstream PV panel tends to be invariant [33]. This is, more or less, similar to the interference effect of the two tandem square cylinders [34,35]. Zhang and Chang [36] investigated the influence of wall shielding effect on wind load of PV arrays near the ground, which implies that the surface wind pressure and bending moment coefficient of the PV panel arrays are strongly tied with the height of the wall. It was shown that the PV modules in the front rows are more sensitive to wind loads than that on the rear edge, and increase the height of parapet can significantly reduce the wind suction on PV panels on flat roof. Such connection, however, tends to vanish when the parapet height exceeds a threshold value.

On the other hand, the wind load and wind-induced response of PV arrays are also dependent on the surroundings. From a practical point of view, oftentimes, the PV arrays are installed on the building roof [37,38], (as shown in Figures 6 and 7). On this account, the wind load on PV panels can be heavily affected by the configuration of the building [39–41]. Schellenberg et al. [42] concluded that rooftop solar PV panel arrays exhibited complex structural responses to wind loads due to the nonlinear behavior of friction and uplift on the roof surface, as well as the complexity of load distribution among PV panel models. Kopp et al. [43] installed two PV arrays on buildings with different angles, aiming at examining the relationship between the aerodynamic effect of the PV array and the building. The results revealed two main mechanisms of aerodynamic load: (1) for high tilt angles, the significant turbulence generated by the array tends to increase the net wind load; (2) for low tilt angles, by contrast, the wind load is mainly dominated by pressure equalization. Note that, compared to ground-mounted system, the existence of building might considerably alter the wind loads due to the complex interactions between building-generated vortices and array-induced flows. Cao et al. [44] reported that the turbulence generated by the PV panel becomes increasingly evident and the pressure equilibrium is weakened as the tilt angle increases. When the panel-to-panel distance increases, the panel-derived turbulence increases. However, the effect of building depth and parapet height on negative modulus force was much less significant. Wang et al. [45] conducted detailed investigation on the effect of building side ratio, aspect ratio and parapet height on wind pressure distribution and mean wind pressure coefficient of PV array. It was shown that the negative critical value of the mean wind pressure coefficient under each wind direction is enhanced with increasing side ratio, while the positive critical value of the mean wind pressure coefficient remains somewhat invariant. On the other hand, the critical value of mean wind pressure coefficient exhibits a negative correlation with the aspect ratio, i.e., the increase of the aspect ratio results in the decrease of the critical value of the mean wind pressure coefficient. Concerning the effect of parapet height, the positive critical value of mean wind pressure coefficient decreases with the increase of the parapet height. The negative critical value of the mean wind pressure coefficient of the building with the parapet is smaller in magnitude than that of the building without the parapet. Alrawashdeh and Stathopoulos [46] highlighted the influence of geometric scale on wind-induced pressure on rooftop solar panels.

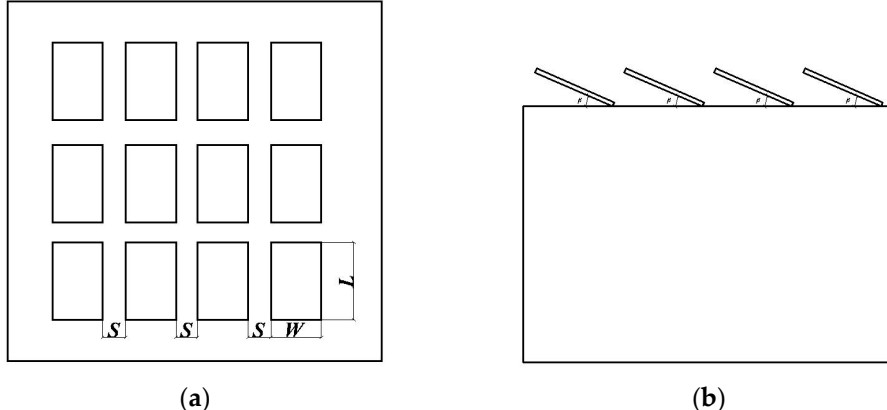

**Figure 6.** Definition of PV panel arrays mounted on flat roofs: (**a**) Horizontal view; (**b**) Vertical view.

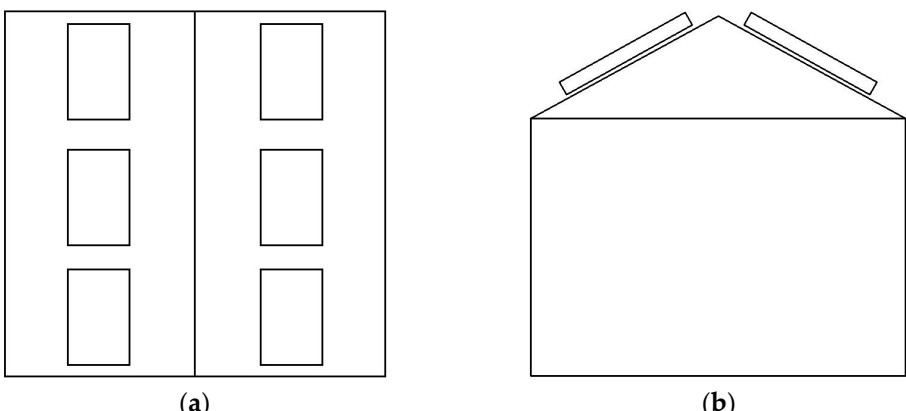

**Figure 7.** Definition of PV panel arrays mounted parallel to roofs: (**a**) Horizontal view; (**b**) Vertical view.

## 2.2. Numerical Simulation

Although, wind tunnel test is an industry wide adopted tool, and has often been considered the principle source of information for wind load estimation and codification, it can be sometimes costly and time-consuming. Also, from a practical point of view, the majority of the existing boundary layer wind tunnel was designed for testing models with a length scale spanning from 1:500 to 1:100. Reproducing the representative aerodynamic models of the PV panels at such scales makes the model too small, which unavoidably cause several technical difficulties. For instance, when surface pressure measurement was adopted to characterize the wind load on PV panels, the resolution of the pressure taps becomes too small, and uncertainty in the flow measurement speed can be high. On the other hand, testing PV panels with enough resolution at relatively larger scales (1:30 and less) can be difficult with respect to the modeling of the lowest 10 m of the atmospheric boundary layer. With the rapid development in computational ability, numerical simulation technique, i.e., Computational Fluid Dynamics, are receiving increasing interest to effectively diagnose the wind-induced effect on high-rise buildings [47–52]. The reliability of numerical simulation was validated by comparing with its counterpart in experimental measurement [47].

With the usage of numerical simulation, the effect of array spacing, ground height and inflow wind direction on the PV panel array have been examined in several previous studies. Su et al. [52] used CFD numerical simulation to assess the wind load on solar panels. The results showed that streamwise distribution of mean surface wind pressure on a PV panel, were overall consistent with that of wind tunnel test. In addition, the connection between wind uplift and aspect ratios, the effect of inclined angle and clearance of a PV panel were likewise diagnosed. It is found that wind uplift of a solar panel increases when there is an increase in inclined angle and the clearance above ground shows an opposite

effect. Meroney and Neff [53] investigated the wind effects on solar PV arrays where both CFD and wind tunnel evaluation were performed. For CFD numerical simulation, both k-ε, RNG and k-ω turbulence model were adopted to predict wind loads. It was evidenced that, the k-omega and RNG are more preferred in terms of modeling the wind load as their results exhibit satisfactory agreements with that in wind tunnel experiment. By means of finite volume method, Reina and De Stefano [14] focused on the mean wind load of the solar-tracking ground-mounted PV panel array. The dynamic mesh model was applied to simulate the wind load under continuous rotation. Evidences have also shown that with different turbulence intensities, the aerodynamic coefficients of PV panel arrays might also change [54]. The aerodynamic coefficients are maximum at the first row of the PV panel arrays, and decreases towards the downstream. The drag and lift force of PV panels are enhanced with the increase of the turbulent kinetic energy, particularly for the first row of panels. As for the effect of inflow wind direction, larger lift and drag coefficients are most likely to occur when the flow comes from 0° and 180°. Jubayer and Hangan [4,55] carried out a systematic study on the flow field around an independent PV system with an inclination angle of 25°. The static aerodynamic coefficients of the PV system were determined, and the velocity, vorticity and wind pressure around the PV panel were discussed.

In wind engineering community, both RANS and large eddy simulation (LES) method are widely used numerical simulation techniques. The RANS method is mainly based on time average, while the large eddy simulation method is based on spatial average, which often enables better capture of the pulsation information of wind load [56–65]. Considering the use of RANS model on PV system, Jubayer and Hangan [62] applied 3D RANS simulations to examine the wind load and flow field around ground-mounted PV system with 25° panel tilt angle, in which the unsteady solver with steady inlet condition and the shear stress transport (SST) k-ω turbulence model were used. Li et al. [63] delivered a detailed assessment on the surface pressure distributions and flow field on PV panel arrays using the RANS method. It was noted that RANS simulation depends strongly on inflow wind direction. Therefore, the performance of several RANS turbulence models was compared. The influence of wind direction, tilt angle and roof clearance were discussed. Fogaing et al. [64] adopted both steady and unsteady RANS models to simulate the near-wake of an array of PV modules. The effectiveness and accuracy of RANS- k-ε, RANS- k-ω (SST), Reynolds Normalization Group (RNG) k-epsilon and Reynolds Stress Model (RSM) were compared with traditional Direct Numerical Simulations (DNS). It was shown that all of the considered models appear to over-predict the mean recirculation length, while under-predict the mean drag coefficient. Irtaza et al. [65] adopted RANS method with unsteady solver to investigate the 3D 1:50 scaled PV panel model subjected to turbulent winds.

Likewise, from a practical standpoint, the LES method was adopted by Wang [58,59] to assess the flow characteristics of PV arrays installed on flat-roofed buildings under various inflow wind direction, the computational domain and grid discretization were shown above the picture (See Figure 8). The mean and transient flow field around the PV panel arrays was detailed, and their connection with wind pressure distribution was discussed. Aly and Aly [66] carried out intensive CFD numerical simulation method on PV panels with the emphasis on understanding the model scale and the inflow effects on the pressure distribution. The CFD was performed using LES and compared with wind tunnel test results. The results revealed that LES is capable to adequately capture both mean and peak pressure fluctuations. Choi et al. [56] applied LES to estimate the wind loads on solar arrays at a series of turbulence intensities and wind speeds.

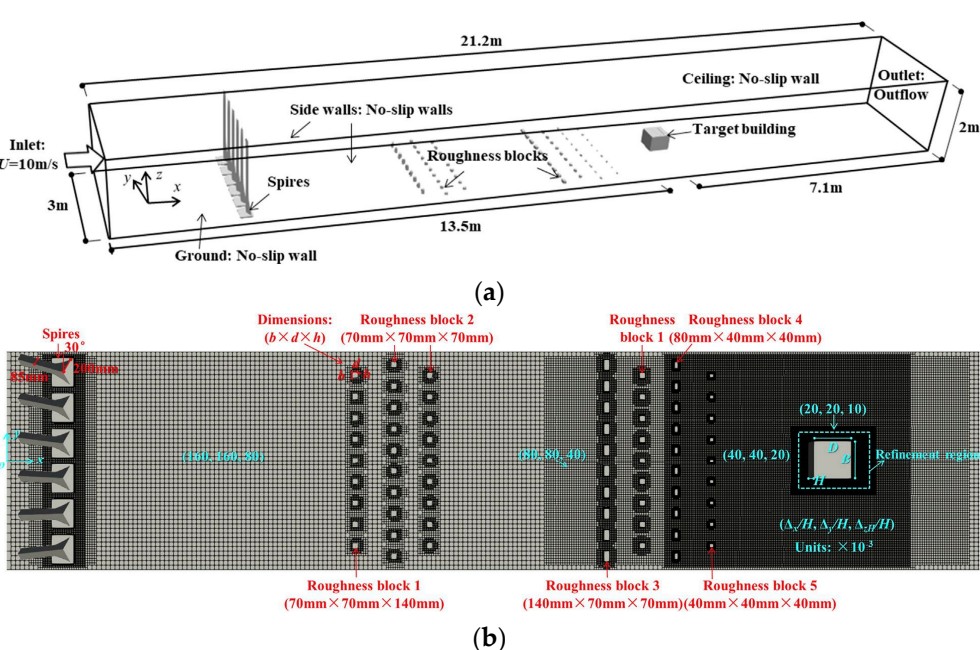

**Figure 8.** Schematic diagram of computation domain for LES: (**a**) Computational domain and boundary conditions; (**b**) Computational mesh [59].

However, it is worth mentioning that existing CFD numerical simulation on wind-induced effect on PV system have focused primarily on traditional roof- or ground-mounted type PV systems. Studies focusing on flexible PV system and its supporting structures are, comparatively, much limited.

## 3. Dynamic Response of Flexible PV Support Structures

Over the past few decades, wind tunnel tests and CFD numerical simulations have been carried out extensively, with the emphasis on diagnosing the dynamic response of various structures, such as cylinders and bridges [67–71], and flexible structures, such as suspension bridges, cable-stayed bridges, solar wing systems and flexible PV systems are usually highly vulnerable to wind-induced vibration [72,73]. It is to be noted that, the wind-induced effect on flexible structures can be divided into two categories: static action and dynamic action. Under certain circumstances, the load induced by wind can damage or destroy the structures. It is therefore of essential importance to better comprehend the dynamic response of the flexible structures.

### 3.1. Buffeting

Buffeting represents a random forced vibration response due to the random fluctuating wind. Buffeting also represents a major component of the wind-induced dynamic response of plate structures with high frequency [73–75], which shows negative impact on the safety of local components in structural construction. Davenport [76] first applied the theoretical method of wing buffeting to bridge structure, and used probability statistic method to investigate wind-induced vibration of bridge. In the analysis of buffeting, the fluctuating wind is often considered as a stationary random process with zero mean. However, in real situation, the wind is most likely to be non-stationary, and thus the structural buffeting response is non-stationary [77]. Using numerical simulation technique, He et al. [78] analyzed the influence of non-stationary and stationary winds on the dynamic response of the vehicle-bridge system. It was shown that the acceleration at the mid-span of the cable-stayed bridge tends to increase when subjected to non-stationary transverse wind.

In addition to the non-stationarity of wind, the non-Gaussianity of wind is also crucial for buffeting analysis [79,80]. Cui et al. [81] reported the existence of non-Gaussian turbulence in the atmosphere. The results suggested that traditional Gaussian assumption

can be non-conservative when analyzing the vibrations caused by non-Gaussion turbulence. Previous studies have indicated that the skewness of non-Gaussian turbulence is strongly tied with the extreme response. Xu et al. [82] found that the lateral and vertical buffeting displacements exhibit oblique distribution. In comparison with Gaussian wind hypothesis, the non-Gaussian wind tends to reduce the lateral extreme response, but increases the vertical extreme response. Da Costa et al. [83] underlined the influence of inclined widen direction on turbulence and motion-related forces. Li et al. [84] determined the root-mean-square (RMS) of chattering lift force and displacement of fairshaped bridges as a function of turbulence integral scale, bridge half-width (γ)ratio and mean wind speed, in which the dependence of the chattering response of large-span suspension bridge on turbulence integral scale was clearly revealed. Xu et al. [85] focused on the chattering response of standard girder segment model of Sutong Bridge. The numerical simulation results showed that the wind-induced vibration of the model in high-level turbulence wind is governed by inflow turbulence, and the characteristic turbulence shows little effect on the chattering response of the model.

### 3.2. Flutter

Flutter describes the oscillation phenomenon due to aerodynamic instability and parameter uncertainty [86–88]. Scanlan and Tomko [86] first extended the concept of flutter derivative in aviation industry, and established a linear flutter theory that applicable to the section of bridge. Such theory, however, is not suitable for large amplitude vibration. Ge and Tanaka [89] presented a comprehensive review on the bridge flutter analysis, and compared the multi-mode and full-mode calculation methods by examining the aerodynamic flutter of long-span suspension bridges. Wei et al. [90], by means of experimental analysis, found that wind-resistant cables can effectively control the flutter stability of long-span suspension bridges. Understanding the critical wind speed is very important to grasp the flutter performance of flexible structures. In the wind tunnel experiment of elastic suspension segmental models, Li et al. [27] studied the flutter performance of the flexible PV support and found that setting central stabilizer plate could not effectively improve the flutter critical wind speed, and of which the flexible PV support would firstly decrease and then increase with the increase of the inclination in different ranges. Wang and Wu et al. [91,92] examined the flutter derivatives of thin plates under various inflow wind direction, and determined the critical wind speed by coupling flutter analysis and free-vibration test. Huang et al. [93] identified eight flutter derivatives of two types of cantilevered wing slab box girder sections, and obtained their respective critical flutter wind speed using Scanlan's two-dimensional flutter theory.

### 3.3. Vortex-Induced Vibration

When the structure is positioned in a flow at a certain wind speed, pairs of alternating shedding vortexes are to be formed on both sides of the structure, which can lead to structural vibration. The VIV (vortex-induced vibration), in comparison with other types of wind-induced vibration, is less likely to cause remarkable damage. The VIV exhibits two typical characteristics: (1) the resonant critical wind speed is relatively low [94]; (2) the oscillation amplitude is small [95–97]. VIV can lead to structural fatigue damage, and influence the comfort level and safety of structures [98]. Based on a series of wind tunnel experiment, He et al. [25] examined the WIV (wind-induced vibration) characteristics of PV modules supported by the suspension cables with high flexibility and low damping ratio. It was found that larger vibration often occurs under cross-wind, and the vibration amplitude indicates a positive correlation with wind speed. It was also reported that torsional vibration is much more pronounced than the vertical vibration. Xie and Fan [99] focused on the natural vibration characteristics of the cable of the pre-stressed flexible PV support system, as well as wind-induced vibration. It shows that, when adding the pre-stressing force, the natural frequency of the structure can be staggered with the wind load pulsation frequency, which can avoid resonance and reduce the wind-induced vibration

of the structure. Kim and Tamura et al. [72] examined the aeroelastic instability of a solar wing system by means of scaled model test. The considered solar wing system consists of 12 PV panels, supported by two cables. Two different span ratios were adopted, and 18 wind speeds were used. The results showed that, with a span ratio of 2%, the vertical displacements exhibit a monotonic increase with mean wind speed, revealing no evident vibrations when the wind direction lies between 0–30°. A sudden enhancement in fluctuating displacement tends to occur with the wind speed is about 10 m/s at the wind direction of 40°. By contrast, with a span ratio of 5%, more complicated vibration in fluctuating displacement at lower wind speeds was obtained. Liu et al. [100] investigated on the wind-induced and critical wind speed of a 33-m-span flexible PV support structure by means of wind tunnel test on the elastic model. The effectiveness of three different types of stability cables on enhancing the critical wind speed of the flexible PV support structure was assessed. It was shown that, 0° and 180° were the two least desirable inflow wind direction, which often results in larger wind-induced response. The critical wind speed is about 18.5 m/s, and the threshold displacement is about 1/100 of the span width. More importantly, with the use of different stability cables, the critical wind speed increased considerably, with a maximum critical wind speed exceeding 36.7 m/s.

Chen et al. [101] applied the active suction jet control method to suppress the vortex-induced vibration of the single box beam, and used time-resolved particle image velocimetry (PIV) to characterize the surrounding wind field. The results were clear, that active suction and spray slit exhibit significant inhibitory effect on the amplitude of vertical and torsional vortex-induced vibration. Li et al. [102] examined the effectiveness of a series of vortex aerodynamic optimization measures, and reported that installing air nozzles on both sides of the side main beam is most useful for suppressing vortex-induced vibration. Tamimi et al. [103] focused on the response of an oscillating square cylinder in the wake of a fixed circular cylinder with three different spacing ratios. The results showed that the vortex-induced vibration characteristics occurred on the downstream square cylinder, which is related strongly to the formation of vortices in the wake of the upstream fixed circular cylinder. The obtained pressure, lift force and displacement for the downstream cylinder were marked different from those of the upstream cylinder. This evidences the interference effect on vortex-induced vibration of the cylinder structure.

Moreover, for flexible structures, numerical simulations have been more frequently adopted. It was shown that, Wang et al. [104,105] used finite element simulation to investigate the wind-induced vibration of multi-row large-span flexible PV supports, where torsion and shaking phenomenon of the flexible PV supports were observed. Similarly, Du et al. [26] used ANSYS to analyze the wind vibration response of long-span flexible PV supports. The results indicated that the vertical displacement response is much sensitive to wind speed than that of the downwind displacement response. Ryan et al. [106] focused on the effect of mass ratio of the structure, in which the critical mass ratio is found to vary between Re = 40~95. More specifically, the critical mass ratio decreases with increasing Reynolds number. Chen et al. [107] analyzed the VIV characteristics of three tandem circular cylinders with different spacing ratios (L/D = 1.2~5.0). The results indicated that the vibration response of the circular cylinders is a function of the spacing ratio.

It is well acknowledged that the bridge cable structure is sensitive to VIV due to its large slenderness ratio, light weight and low natural frequency. Existing literature has shown that, when the span length of bridge increases, the frequency and slenderness ratio of the cable reduces, which might result in various new issues of wind-induced vibration [108,109]. However, studies targeted on the wind-induced responses of flexible PV systems are somewhat limited, which should be further examined.

## 4. Conclusions

This study focused specifically on the state-of-the-art researches on the aerodynamic characteristics and wind-induced response of flexible PV systems. The major concluding remarks are summarized as follows:

1. The aerodynamic characteristics and wind-induced response of the flexible PV system can be affected by a series of governing factors, such as inclination, inter-plate spacing, wind direction angle, installation position and surrounding environment.

2. Regarding the investigation of flexible PV system, both wind tunnel and numerical simulation techniques are considered to be useful. Wind tunnel test is industry standard tool, but it might subject to scale issue. Numerical simulation, on the other hand, is less costly and more efficient.

3. As a cable-plate structure system with high flexible and nonlinear vibration, the flexible PV system exhibits pronounced aerodynamic characteristics, such as chattering, vortex, flutter and wake surge responses. However, relevant studies remain limited, which needs to be further considered.

**Author Contributions:** Conceptualization, F.C. and Y.Z.; methodology, Y.Z.; software, Y.Z.; validation, F.C., Y.Z. and W.W.; formal analysis, Z.S.; investigation, W.W.; resources, F.C.; data curation, Y.Z.; writing—original draft preparation, Y.Z.; writing—review and editing, Y.Z. and Y.L.; supervision, Yi Li; project administration, F.C.; funding acquisition, F.C. All authors have read and agreed to the published version of the manuscript.

**Funding:** This project was funded by National Natural Science Foundation of China (Project No: 52278479), Open Foundation of Key Laboratory of Safety and Control for Bridge Engineering of CSUST, Ministry of Education (13KB01).

**Acknowledgments:** Thanks to the anonymous reviewers and the editors for their valuable comments and suggestions.

**Conflicts of Interest:** The authors declare no conflict of interest.

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
