# Peer review of "A Review on Aerodynamic Characteristics and Wind-Induced Response of Flexible Support Photovoltaic System"

_atmosphere, doi:10.3390/atmos14040731_

Round 1

Reviewer 1 Report

Based on photovoltaic system, this paper summarizes the wind resistance research process of photovoltaic system in recent years, emphasizes the importance of flexible photovoltaic system in the future, and points out the direction for the future study of dynamic response of flexible photovoltaic system. The conclusion is simple and easy to understand, and the object of study has a wide range of practical application value. Some suggestions are listed as follows:

1. Some formats of the references in this review are wrong and should be corrected.

2. The word of this manuscript seems a little short, which is suggested to increase the number of words appropriately, enrich the content of the review.

3. In the third part of “Research progress on dynamic response of flexible structures”, it is suggested to refine the language again and improve the writing ability.

Author Response

Please refer to the response in the attached file.

Reviewer 2 Report

This review focuses on the flexible photovoltaic(PV) system which has been widely concerned at present, offering a systematic summarize on the wind-induced response and aerodynamic characteristics of flexible PV system in recent years by means of experiments and numerical simulation methods, and emphasize the importance of studying dynamic response to the flexible PV system by combining with high flexible structures such as bridges and high-rise buildings, which is of great reference value for future research on flexible PV system, and contributes to the promotion and development of solar energy as a renewable energy source. However, in the meantime, the review also has the following problems:

1. Some references in this review are formatted incorrectly and need to be corrected.

2. In the Section of “2.2 Numerical simulation”, RANS and LES methods should be introduced separately to make the structure of the article neat and tidy.

3. There are few pictures in the review, so the picture description can be added appropriately.

Author Response

For response, please refer to the attached file.

Reviewer 3 Report

The review, "A review on aerodynamic characteristic and wind-induced response of flexible photovoltaic system" is an interesting and well-structured paper. My critique is that the english language in the paper needs to be improved. For example several sentences (including the first one of the abstract) are too long. As another example, many articles, such as "a" and "the" are missing.

Author Response

For response, please refer to the attached file

Reviewer 4 Report

It is a review work on the wind issue of the flexible photovoltaic system, which is get more and more widely used. As a review, plenty of pictures  introducing the development of the work will be expected. However, it is not seen in the paper. The introduction for wind loading and wind-induced response of other photovoltaic system and other flexible structure can be found in the paper. That is to say, the review is not focused on the detailed information of the flexible photovoltaic system. For the wind-induced response of the flexible PV system, the buffeting and VIV problem are the main points compared with the traditional PV system and should be well addressed. Also, several damages of the flexible PV system due to the severe wind have been reported and should be introduced.

Author Response

(The authors gave the same response as above.)

Round 2

Reviewer 4 Report

The paper is well revised and the comments are all addressed. It can be published after minor modification. Here are some comments.

1. Some of the the citation of the reference in the paper may be incorrect. For example, et al. is missing for a paper written by a team. Another example, wenyong[25] should be cited as Ma et al. [25].  A thorough check is needed for the whole paper.

2. The wind tunnel testing examples and some testing results of the flexible PV should be provided because the title of the paper is the flexible PV. And in the paper, only the picture of the roof PV wind tunnel testing cases is presented. 

 3. According to the paper, the subtitle of 3.1 is dynamic response, 3.2 is buffeting, 3.3 is flutter, 3.4 is VIV. However, this classification may not be suitable because the buffet, flutter and VIV are all dynamic response. And the title of the Section 3 is Dynamic Response. A reorganization of the section 3 is needed. 

Author Response

Dear reviewer, enclosed please find our responses. 
